# Developments in Genetics: Better Management of Ovarian Cancer Patients

**DOI:** 10.3390/ijms242115987

**Published:** 2023-11-05

**Authors:** Ovidiu-Virgil Maioru, Viorica-Elena Radoi, Madalin-Codrut Coman, Iulian-Andrei Hotinceanu, Andra Dan, Anca-Elena Eftenoiu, Livia-Mălina Burtavel, Laurentiu-Camil Bohiltea, Emilia-Maria Severin

**Affiliations:** 1Department of Medical Genetics, “Carol Davila” University of Medicine and Pharmacy, 020021 Bucharest, Romania; maioru.ovidiu@gmail.com (O.-V.M.); mcoman29@gmail.com (M.-C.C.); dandra.crgm@gmail.com (A.D.); anca.eftenoiu@gmail.com (A.-E.E.); burtavelmalina@gmail.com (L.-M.B.); laurentiu.bohiltea@umfcd.ro (L.-C.B.); emilia.severin@umfcd.ro (E.-M.S.); 2“Alessandrescu-Rusescu” National Institute for Maternal and Child Health, 20382 Bucharest, Romania

**Keywords:** ovarian, molecular, genetics, pathways, diagnosis, screening, panel

## Abstract

The purpose of this article is to highlight the new advancements in molecular and diagnostic genetic testing and to properly classify all ovarian cancers. In this article, we address statistics, histopathological classification, molecular pathways implicated in ovarian cancer, genetic screening panels, details about the genes, and also candidate genes. We hope to bring new information to the medical field so as to better prevent and diagnose ovarian cancer.

## 1. Introduction

Ovarian cancer occupies a significant position within the landscape of oncological diseases, captivating researchers and medical professionals alike. Its prominence can be attributed to a combination of factors that contribute to its scientific complexity and societal relevance. One of the primary facets that renders ovarian cancer intriguing is its multifaceted etiology, which involves intricate interactions between genetic predispositions, hormonal influences, and environmental factors. The diversity of clinical presentations exhibited by ovarian cancer further deepens the intrigue, as the disease often eludes early detection and diagnosis due to subtle or non-specific symptoms, necessitating a comprehensive understanding of its varied manifestations. Moreover, the global impact of ovarian cancer on morbidity and mortality lends urgency to its investigation. The disease’s prevalence, coupled with its relatively high fatality rates, underscore the imperative for thorough scientific exploration. The toll it imposes on individuals in its grip, along with their loved ones and healthcare systems, highlight the urgent requirement for progress in prevention, early detection, and treatment methodologies.

## 2. Ovarian Cancer Statistics

According to the GLOBOCAN 2020 database of the International Agency for Research on Cancer (IARC), ovarian cancer is the eighth most common type of cancer in women and the seventh most common cause of cancer death in women. The database in 2020 estimated a worldwide age-standardized incidence rate of ovary cancer of 6.6 per 100,000 and a worldwide age-standardized mortality rate of 4.2 per 100,000 [1].

In accordance with the prognostications in GLOBOCAN 2020 projections, it is anticipated that by the year 2040, the global incidence of ovarian cancer among women will exhibit a notable escalation of approximately 42%, resulting in a total of 445,721 new diagnoses. Moreover, the annual toll of fatalities attributed to ovarian cancer is poised to undergo a significant augmentation, surging by over 50% in comparison to the statistics of 2020 and, reaching a projected count of 313,617 deaths.

The survival rates over a five-year span subsequent to an ovarian cancer diagnosis exhibit noteworthy disparities across various nations. For instance, within more developed nations, the existing survival rates fluctuate within the range of 36% to 46%. Nonetheless, in certain countries, this metric assumes notably diminished values. Collectively, these survival rates consistently lag behind those observed in other malignancies, such as breast cancer, wherein numerous countries record five-year survival rates that approach the vicinity of 90% [1,2].

The estimated incidence in the U.S. is 10.3 per 100,000 per year, and the estimated mortality rate is 6.3 per 100,000 per year (age adjusted and based on 2016–2020 data) [3].

The projections provided by the American Cancer Society regarding ovarian cancer within the United States for the year 2023 depict the following statistics:

Approximately 19,710 women will be newly diagnosed with ovarian cancer.

An estimated 13,270 women will succumb to ovarian cancer.

Ovarian cancer assumes the fifth position in the hierarchy of cancer-related fatalities among women, surpassing all other malignancies originating from the female reproductive system. The probability of an individual of the female gender developing ovarian cancer throughout her lifespan is estimated to be roughly 1 in 78. Simultaneously, the likelihood of her experiencing mortality attributable to ovarian cancer stands at approximately 1 in 108 [4].

As far as Romania is concerned, according to GLOBOCAN 2020, ovary cancer was the seventh most common type of cancer in women and the fifth most common cause of death from cancer in women. The estimates for ovarian cancer in Romania were 10.6 per 100,000 for age-standardized incidence and 5.3 per 100,000 for age-standardized mortality [1].

Ovarian cancer has the worst prognosis among gynecological cancers. The U.S. National Cancer Institute estimates a 5-year relative survival rate for ovarian cancer of 50.8% based on 2013–2019 data compared to 81.0% for uterine corpus cancer, 69.9% for vulvar cancer, 67.2% for cervical cancer, and 53.2% for vaginal cancer [3].

### Ovarian Cancer Risk Factors

Genetic predisposition plays a pivotal role in the risk factors associated with ovarian cancer [5,6]. Ovarian cancer has been associated with penetrant germline pathogenic mutations in tumor suppressor genes, especially BRCA1 and BRCA2 [7]. Other causes of hereditary ovarian cancer are represented by Lynch syndrome (mutations of mismatch repair genes) and Li–Fraumeni syndrome (TP53 mutation) [8]. Mutations in other genes involved in DNA repair, such as PALB2, RAD51, and CHEK2 may be expressed in ovarian cancers [8]. Inheriting mutations in these genes can substantially elevate the risk of developing ovarian cancer, often at an earlier age.

Risk factors for ovarian cancer include advanced age, obesity, pregnancies after 35 years old or nulliparity, hormone replacement therapy, and family predisposition [5,6]. Ovarian cancers can be hereditary (familial) or sporadic (non-hereditary).

Environmental factors also exert their influence on ovarian cancer risk. Exposure to certain substances, such as talcum powder or asbestos, has been linked to an elevated risk of ovarian cancer. Additionally, hormonal factors, including early onset of menstruation, late onset of menopause, and the use of hormone replacement therapy, can impact the hormonal balance within the body and subsequently influence ovarian cancer risk.

Lifestyle factors further compound the complexity of ovarian cancer risk. Obesity, for instance, has been associated with increased levels of chronic inflammation and altered hormonal profiles, potentially contributing to the development of ovarian cancer. Conversely, regular physical activity and a balanced diet rich in antioxidants have been suggested to mitigate ovarian cancer risk by promoting overall health and reducing oxidative stress.

## 3. Histopathologic Classification of Ovarian Cancer

Ovarian tumors exhibit a diverse array of histological characteristics and genetic profiles, prompting the need for a comprehensive classification system. This section delves into the classification of primary ovarian tumors and highlights their specific genetic patterns, shedding light on the intricate interplay between morphology and underlying genetic alterations [9].

A.Epithelial Tumors

Epithelial ovarian tumors, the most common type, encompass various subtypes, each with distinct genetic markers. Ovarian cancer is widely acknowledged as the most fatal among gynecologic malignancies, causing a significant impact on global health. The spectrum of epithelial ovarian cancer encompasses a diverse range of subtypes, consisting of five primary categories: high-grade serous ovarian carcinoma (HGSOC), low-grade serous ovarian carcinoma (LGSOC), endometrioid ovarian cancer (EnOC), ovarian clear cell carcinoma (OCCC), and mucinous carcinoma. The prevalent subtype, HGSOC, constitutes the majority, encompassing approximately 70% of all cases. Serous tumors, which include serous cystadenoma, adenofibroma, and surface papilloma, often carry genetic mutations in the BRCA1 and BRCA2 genes. Low-grade serous carcinoma and high-grade serous carcinoma are linked to aggressive malignancy and may exhibit BRCA, BRAF, and KRAS mutations. Similarly, mucinous tumors, like mucinous cystadenoma and adenofibroma, exhibit diverse genetic profiles, while endometrioid tumors, including endometrioid carcinoma, may be associated with Lynch syndrome and microsatellite instability, shedding light on their underlying genetic predispositions.

Four gene mutations are most commonly reported to be highly associated with epithelial ovarian cancer, including: TP53, BRCA1/2, PIK3CA, and KRAS. The frequency of these mutations varies among different subtypes of epithelial OC (Table 1) [10]. 

The expression of P53 mutation is the most frequently encountered mutation in HGSOC. The P53 mutation rate increases to 54.5% in HGSOC. On the other hand, the majority of hereditary OC cases are determined by BRCA1/2 genes. The BRCA mutation rate increases to 40% in recurrent HGSOC. PIK3CA mutations are reported with high frequency in OCCC and in EnOC in relation to endometriosis. The presence of the KRAS mutation is a significant factor in the development of LGSOC and mucinous OC. The possible connections between these mutations and OC involve several mechanisms, including the disruption of genes responsible for tumor suppression, defects in DNA repair genes, alterations in apoptosis, activation of oncogenes, and epigenetic silencing [10,11,12].

B.Mesenchymal Tumors

Mesenchymal tumors, originating from ovarian stromal tissue, offer unique genetic insights. Endometrioid stromal sarcoma, characterized by both low and high grades, may present genetic alterations contributing to its aggressive behavior. Leiomyomas, smooth muscle tumors, and ovarian myxomas could also harbor distinctive genetic patterns that contribute to their development. MED12 mutations are the most common genetic modifications in benign leiomyoma, although they are far less common in its malignant counterpart, leiomyosarcoma.

SLITRK4, OSTN, MASP1, NLGN4X, NLGN1, XRN2, ASS1, TSPAN7, RORB, and HRASLS were among the genes shown to be overexpressed in primary Uterine Leiomyosarcomas. Genes like TNNT1, FOLR3, TDO2, CRYM, GJA1, TSPAN10, THBS1, SGK1, SHMT1, EGR2, and AGT were shown to be overexpressed in Uterine Leiomyosarcomas metastases [13,14].

C.Sex Cord Stromal Tumors

Sex cord stromal tumors, arising from specialized cells within the ovaries, showcase a spectrum of genetic alterations. Adult granulosa cell tumors, for instance, may carry mutations in the FOXL2 gene. Sertoli–Leydig cell tumors, including well-, moderately, and poorly differentiated forms, might exhibit unique genetic signatures driving their diverse clinical behavior. Sertoli–Leydig cell tumors of the ovary are rare tumors predominantly caused by mutations in the DICER1 gene. The majority of cases are caused by autosomal dominant germ line mutations in the serine/threonine kinase 11/liver kinase B1 (STK11/LKB1) gene on chromosome 19p13 [15,16]. 

D.Germ Cell Tumors

Germ cell tumors encompass an array of genetic backgrounds. Dysgerminoma, a germ cell tumor with high sensitivity to chemotherapy, can display amplifications in the 12p region. Yolk sac tumors, aggressive malignancies often seen in younger patients, may exhibit aneuploidy, loss of parts of 4q and 6q, and gain of parts of the 20q region. Teratomas, comprising various tissue types, could present complex genetic landscapes. KRAS, CDKN1B, CCND2, ETV6, and RAD52 mutations are the most common in germ cell tumors. The most prevalent changes in germ cell tumors are CDKN1B Amplification, KRAS Amplification, CCND2 Amplification, ETV6 Amplification, and RAD52 Amplification [17,18,19].

E.Miscellaneous and Tumor-Like Lesions

Tumors falling under the miscellaneous category, such as solid pseudopapillary tumors and small cell carcinomas, might be driven by unique genetic alterations. Additionally, non-neoplastic tumor-like lesions, including corpus luteum cysts and hyperreactio luteinalis, could exhibit distinct genetic patterns associated with their formation. Miscellaneous ovarian tumors include rete ovarii tumors, mesothelial tumors, lymphoid and myeloid tumors, soft tissue tumours, and different tumor-like abnormalities.

Aligned with the criteria set forth by the World Health Organization (WHO), this exploration encompasses a range of subtypes, including Mixed Epithelial and Mesenchymal Tumors, like Adenosarcoma, as well as tumor-like lesions, such as Follicle cysts, Corpus luteum cysts, Large solitary luteinized follicle cysts, Hyperreactio luteinalis, Pregnancy luteomas, Stromal hyperplasia and hyperthecosis, Fibromatosis, massive edema, and Leydig cell hyperplasia. Understanding the genetic correlations associated with each of these distinct classifications not only contributes to their accurate identification but also sheds light on their underlying molecular dynamics and potential clinical implications.

The classification of ovarian tumors not only aids in clinical diagnosis but also offers a window into the underlying genetic events driving tumorigenesis. Genetic mutations, alterations, and patterns provide crucial insights into tumor behavior, prognosis, and potential therapeutic strategies. Understanding these genetic nuances within the framework of histopathological classification paves the way for personalized treatments and improved patient outcomes.

Ovarian tumors span a wide spectrum of histopathological subtypes, each intricately linked to a specific genetic background. This section underscores the critical importance of integrating genetic patterns into the classification of ovarian tumors, thereby enhancing our comprehension of their underlying biology and opening avenues for targeted therapeutic interventions.

## 4. Genetic Causes of Ovarian Cancer

Between 5 and 15% or approximately 10% of ovarian cancers are determined by germline mutations in different genes with activity in oncogenesis processes. Among these, the most frequently involved are the BRCA1 and BRCA2 genes, but there are also other well-known ones: CHEK2, BARD1, PALB2, RAD51, and TP53.

BRCA1 and BRCA2

BRCA1 (BReast CAncer gene 1) and BRCA2 (BReast CAncer gene 2) are tumor suppressor genes that play a role in genomic stability and DNA repair. BRCA1 and BRCA2 are involved in homologous recombination, a key pathway for repairing double-strand DNA breaks [20]. BRCA1 and BRCA2 encode large proteins expressed in several tissues throughout S and G2 stages of the cell cycle [21].

The BRCA1 gene is located on chromosome 17q21 and encodes a protein of 1863 aminoacids. At the N-terminal end of the protein, there is a RING (Really Interesting New Gene) domain, which is required for the interaction with BARD1 (BRCA1 Associated RING Domain protein 1). At the C-terminal end, there are two domains (BRCT) that mediate the interaction with proteins, such as ABRAXAS (BRCA1 A Complex Subunit), CtIP, (C-terminal binding protein 1 interacting protein), and BRIP1 (BRCA1 interacting protein C-terminal helicase 1). The coiled-coil domain, which is important for the interaction with BRCA1 through PALB2, and two nuclear localization signals (NLS), essential for BRCA1 function, can be found in the central region [22]. Most BRCA1 mutations are frameshift insertions or deletions, nonsynonymous truncations, and disruptions of splice site resulting in non-functional proteins [23]. Most frequent mutations are found in the regions that correspond to BRCT and RING domains and in the exons 11-13 that encode NLS. BRCA1 mutations have been associated with breast and ovarian cancers, as well as prostate, pancreas, gastric, and colorectal cancers.

The BRCA2 gene is located on chromosome 13q12.3 and encodes a protein of 3418 amino acids [24]. BRCA2 contains two DNA-binding domains [25]. The central region contains BRC repeats that bind to RAD51 [26]. C-terminus contains two NLS and an additional RAD51 interaction site (TR2) [27].

Most BRCA2 mutations are frameshift insertions, deletions, and nonsense mutations, and they result in premature truncation or non-functional protein [25]. The most commonly mutated site is exon 11, which encodes the BRC repeats [28]. Mutations in the BRCA2 gene predispose to breast, ovarian, and prostate cancer, but they have also been associated with ocular or cutaneous melanoma and gastric, pancreatic, gallbladder, and bile duct cancer [29]. 

BRCA-associated breast cancers have demonstrated a dependence on alternate DNA repair processes through base excision repair, which requires poly (ADP-ribose) polymerase 1 (PARP-1) [30]. BRCA mutations follow an autosomal dominant inheritance pattern.

PALB2

The PALB2 (partner and localizer of BRCA2) gene is located on chromosome 16p12.2 and encodes a protein that interacts with BRCA1 and BRCA2, forming a complex that is crucial for DNA repair through homologous recombination [31]. The coiled-coil domain at the N-terminal end of the protein interacts with BRCA1. The WD40 domain, which interacts with BRCA2, DNA polymerase, RAD51, RAD51C, and the ubiquitin ligase RNF168, is located at the C-terminal end of the protein [32,33]. The ChAM domain, which contributes to the formation of PALB2-BRCA2-RAD51, is found in the middle region [34]. The protein has two DNA-binding domains. Moreover, it interacts directly with RAD51 [35]. Breast, ovarian, and pancreatic cancer have all been linked to heterozygous germline mutations in the PALB2 gene [36]. Biallelic mutations in PALB2 are involved in a subtype of Fanconi anemia [37,38]. 

RAD51

RAD51 is a gene located on chromosome 15q15.1 involved in homologous recombination and DNA repair processes [39]. The gene in question produces a protein that belongs to the RAD51 protein family. RAD51 family members are highly similar to bacterial RecA and Saccharomyces cerevisiae Rad51 [40]. RAD51 protein plays a central role in homologous recombination, a mechanism that ensures accurate repair of DNA double-strand breaks. It acts by assembling into a filamentous structure on single-stranded DNA regions, facilitating the search for and pairing with homologous DNA sequences for repair. This protein interacts with the ssDNA-binding protein RPA, RAD52, BRCA1, and BRCA2 [41]. It has been shown that BRCA2 controls its intracellular location as well as its capacity to bind DNA. Following BRCA2 inactivation, the loss of these regulators may be a crucial event triggering genomic instability. Mutations in RAD51 have been associated with an increased susceptibility to certain types of cancer, particularly breast and ovarian cancer. 

CHEK2

CHEK2 (Checkpoint Kinase 2) gene is located on chromosome 22q12.1, and it encodes a protein involved in cell cycle regulation and DNA repair processes. As a checkpoint kinase, CHEK2 plays a critical role in monitoring DNA integrity and ensuring proper cell division. It functions by inhibiting the entry of the cell into mitosis by stopping it in stage G1 in response to DNA damage. The role of the gene is to alter the intercellular signal in cases of DNA damage, thereby inducing a prompt phosphorylation response. Specifically, in situations where the action of this gene is inactivated, various types of cancer will occur, including breast, ovarian, prostate, and colorectal cancer [42]. The types of cancer that can result from gene mutations are both sporadic and hereditary. The 1100delC mutation variant was observed in Cowden syndrome and Li Fraumeni syndrome [43]. The gene is activated by the phosphorylation of Thr68 by ATM, which produces the dimerization of the gene, giving it the ability to function as a kinase. Subsequently, the gene reacts with phosphatase CDC25, protein kinase Ser/THr NEK6, transcription factor FOXM1, protein p53, and BRCA1 or BRCA2. Alterations in the CHEK2 or TP53 genes have been linked to reduced sensitivity to anthracycline-based chemotherapy in breast cancer patients. Another study in Chinese women with breast cancer demonstrated that H371Y carriers may have a better response to neoadjuvant chemotherapy [44].

The TP53 gene

The TP53 gene, often referred to as p53, is a pivotal tumor suppressor gene that plays a crucial role in maintaining genomic stability and preventing the development of cancer. Its position can be found on the short arm of chromosome 17 (17p13.1).

It serves as a nuclear transcriptional regulator intricately involved in numerous cellular processes. Through its direct interaction with DNA, p53 exerts precise control over the expression of a vast array of target genes, thus upholding cellular homeostasis and safeguarding the integrity of the genome. Notably, p53’s role encompasses multifaceted functions, including activation of DNA repair mechanisms following instances of DNA damage, imposition of cell growth arrest through modulation of the G1/S transition, which allows for the execution of DNA repair processes, and initiation of programmed cell death, or apoptosis, in cases of irreparable DNA damage [45].

Structurally, the p53 protein exhibits four discernible functional domains: an N-terminal domain governing transcriptional activation, a central DNA-binding domain endowed with sequence specificity, a tetramerization domain, and a C-terminal regulatory domain. Beyond its recognized capacity for transcriptional activation, p53 has also been associated with transcriptional repression; however, the binding sites implicated in the regulation of its downregulated target genes remain relatively less characterized.

The induction of p53 activation transpires through a myriad of stimuli, encompassing UV- or gamma-irradiation-induced DNA damage, inappropriate activation of proto-oncogenes, mitogenic signaling cascades, stress elicited within the ribosomal or nucleolar milieu, and instances of hypoxia. Upon attaining an activated state, p53 orchestrates an array of cellular responses contingent upon the specific cellular context. These responses encompass cell cycle arrest, senescence, cellular differentiation, apoptosis, and ferroptosis. The mechanism underlying such varied outcomes resides in p53’s aptitude for stimulating the expression of a diverse spectrum of genes vital for these cellular activities.

TPp53 is known in the transcription of p21, an orchestrator of p53-mediated cell cycle arrest in the G1 phase, which culminates in cellular senescence. In parallel, p53 also triggers the expression of key elements, such as Puma, Bax, and miR-34, which collectively underpin p53’s ability to elicit apoptosis. Intriguingly, recent investigations have unveiled p53’s role in inducing ferroptosis—a specialized form of cell demise characterized by reactive oxygen species—via the activation of SLC7A11, a pivotal component of the cystine/glutamate antiporter [46].

ARID1A gene

ARID1A, also known as AT-rich interactive domain-containing protein 1A, is a tumor suppressor gene located on chromosome 1p36.11. Inactivating mutations in the ARID1A gene, which result in the loss of protein expression, have been found in cases of ovarian cancer, especially clear cell carcinoma [47]. This gene encodes ARID1A protein, a member of the SWI/SNF family involved in transcriptional activation through chromatin remodeling. Chromatin remodeling is a process that regulates the structure and accessibility of DNA, which in turn affects gene expression. Mutations in the ARID1A gene can lead to disruptions in the regulation of genes involved in cell growth, differentiation, and DNA repair, contributing to the development of cancer.

The role of ARID1A in cancer is exercised by the involvement in EZH2 methyltransferase activity, the PI3K/AKT/mTOR pathway, the regulation of p53 targets, DNA damage checkpoints, and the tumor immune response [48].

Other cancers associated with Other cancers associated with mutations that lead to loss-of-function in the ARID1A gene are endometrial cancers, gastric cancers, bladder cancers, hepatocellular cancers, melanomas, colon cancers, and lung cancers [49].

### 4.1. Candidate Genes

Candidate genes in the context of ovarian cancer refer to specific genes that are hypothesized to play a significant role in the development, progression, or susceptibility to ovarian cancer. These genes are identified through various scientific methods and investigations that aim to uncover genetic factors associated with the disease. Candidate genes are selected based on their biological functions, potential relevance to cancer biology, and evidence from research studies indicating their involvement in ovarian cancer [50].

The process of identifying candidate genes involves a combination of genetic, molecular, and clinical research, as it is a dynamic process. Candidate genes are often selected based on their known or suspected functions related to cancer biology. For example, genes involved in cell cycle regulation, DNA repair, apoptosis, and signaling pathways that contribute to tumorigenesis are frequently considered [51].

For example, PTEN is a crucial tumor suppressor gene frequently implicated in ovarian cancer. It regulates cell growth, division, and apoptosis by inhibiting the PI3K-AKT signaling pathway. Loss of PTEN function, often due to mutations or epigenetic silencing, can lead to uncontrolled cell growth and survival. 

PTEN exerts its tumor-suppressive influence through the orchestrated action of two principal domains: the phosphatase domain and the C2 domain. In its capacity as a tumor suppressor, PTEN carries out its regulatory role by means of its 3’-phosphatase activity, which engenders the dephosphorylation of phosphatidylinositol (3,4,5)-trisphosphate (PIP3). This dephosphorylation event culminates in the restraint of AKT activity, thereby orchestrating downstream modulation of the entire signaling cascade. Delving further into the molecular intricacies, the PTEN protein exhibits a dual aptitude: lipid phosphatase activity, which is pivotal in halting cell-cycle progression at the G1/S checkpoint, and protein phosphatase activity, which is instrumental in repressing certain processes, such as focal adhesion assembly, cellular spreading, and migration. Furthermore, this dualistic activity extends to the inhibition of the MAPK signaling pathway, which is prompted by growth factor stimuli [52].

KRAS and BRAF

Mutations in KRAS and BRAF genes are commonly observed in mucinous ovarian carcinomas. These genes are part of the MAPK signaling pathway, which regulates cell growth, differentiation, and survival. Aberrations in KRAS and BRAF contribute to dysregulated cellular processes in ovarian cancer [53].

FOXL2

FOXL2 mutations are frequently found in adult granulosa cell tumors, a rare subtype of ovarian cancer. FOXL2 is a transcription factor involved in ovarian development and folliculogenesis. Mutations in FOXL2 can lead to dysregulated gene expression and cell growth in granulosa cell tumors [54].

CCNE1

Cyclin E1 (CCNE1) is involved in cell cycle regulation and progression. Amplification and overexpression of CCNE1 have been observed in a subset of high-grade serous ovarian carcinomas, contributing to uncontrolled cell proliferation and genomic instability [55].

### 4.2. Prognostic Genes

In the realm of ovarian cancer, prognostic genes are those whose expression levels or genetic alterations are associated with the anticipated clinical outcomes for patients. These genes can provide valuable insights into disease progression, patient survival, and response to treatments. Some genes are considered risk genes, having a negative influence on the clinical outcome, while others are considered protective genes, the presence of which positively influences the evolution of the disease. Several prognostic genes have been identified through extensive research in ovarian cancer.

AP3D1 and LRFN4 are genes associated with increased risk, and elevated expression of these genes is linked to diminished survival rates among individuals with ovarian cancer. The presence of AP3D1 and LRFN4 has been found to contribute to the initiation and progression of various cancers and diseases [56].

Notably, in cases of colorectal cancer, the absence of the optic nerve element prompts evasion of the immune system and an inherent resistance to immunotherapy. Intriguingly, the optic nerve element unexpectedly interacts with AP3D1, thereby preventing the sorting and degradation of palmitoylated IFNGR1 lysosomes. This interaction serves to preserve the integrity of signaling related to interferon gamma and major histocompatibility complex (MHC)-I, which are crucial components of the immune response [57].

DCAF10, FBXO16, PTPN2, SAYSD1, and ZNF426 are genes that confer protection to ovarian cancer patients, with their heightened expression correlating with improved survival rates. These genes exhibit a restraining influence on the development of numerous cancers and diseases [56,57].

Notably, FBXO16 serves as a tumor suppressor, and it functions within the skp1-cullin1-f-box protein complex. This complex targets nuclear β-catenin for proteasome degradation through the 26S proteasome system. Deficiency of FBXO16 results in elevated β-catenin levels, subsequently fostering certain processes, such as cancer cell invasion, tumor proliferation, and epithelial–mesenchymal transition. In the realm of breast cancer, FBXO16 holds potential as a clinical target and prognostic biomarker across diverse molecular subtypes. Additionally, research has indicated that augmenting the count of cytotoxic Tim-3+/CD8+ T cells can enhance effective anti-tumor immunity. PTPN2 emerges as an attractive target for tumor immunotherapy in immune cells, further underscoring its significance [58].

CCDC80, also known as LRP1B (Low-Density Lipoprotein Receptor-Related Protein 1B), is gaining recognition as a candidate gene in ovarian cancer prognosis. LRP1B is a transmembrane receptor involved in cellular processes like endocytosis, cell signaling, and cell adhesion. Recent studies have suggested that CCDC80/LRP1B may function as a tumor suppressor in certain contexts. Downregulation or loss of CCDC80 expression has been associated with multiple cancer types, including ovarian cancer, and it has been linked to poorer outcomes.

In the context of ovarian cancer, CCDC80’s role in mediating cell adhesion and migration is of particular interest. Its downregulation may contribute to enhanced tumor cell invasiveness and metastasis. Studies exploring the clinical relevance of CCDC80 in ovarian cancer patients are underway; they aim to elucidate its impact on disease progression and patient survival [59].

FBXO16, an F-box protein, has garnered attention for its tumor-suppressive properties. F-box proteins are key components of ubiquitin ligase complexes that target specific proteins for degradation, thereby regulating various cellular processes. FBXO16 has been identified as a potential tumor suppressor in various cancer types, including ovarian cancer.

Research has highlighted FBXO16’s involvement in the regulation of β-catenin, a key player in the Wnt signaling pathway that influences cell proliferation, differentiation, and migration. FBXO16 helps target nuclear β-catenin for degradation, thereby curbing its oncogenic potential. Loss of FBXO16 could lead to increased β-catenin levels, contributing to cancer cell invasion, tumor growth, and metastasis.

FBXO16’s significance extends to potential clinical applications. It has been suggested that FBXO16 might serve as a clinical target for therapeutic intervention, particularly in breast cancer, where its role as a prognostic biomarker across distinct molecular subtypes is being explored [60].

## 5. Molecular Pathways in Ovarian Cancer

A.The MAPK signaling molecular pathways

Several signaling pathways involved in ovarian tissue carcinogenesis have been described.

The BRCA1 and BRCA2 pathway—DNA damage repair (strand breaks) and transcription regulation(Figure 1). The BRCA1/2 genes are involved in homologous recombination, the most common repair mechanism used by cells. The signaling pathway also includes other proteins encoded by the ATM, ATR, BLM, CHEK2, CHEK1, ERCC1, MLH1, PALB2, RAD50, and PTEN genes.

B.The MAPK/ERK pathway that acts through the KRAS, MAPK1, NF1, NF2, and BRAF genes (Figure 2).

The MAPK (mitogen activated protein—kinase) pathway is involved in signal transduction, with high oncogenic potential, through two classes of surface receptors:Tyrosine-kinase receptor;G protein-coupled receptor.

Both classes of receptors interact with cytokines, growth factors, etc. The MAPK pathway has important functions in basic cellular activities: cell growth and differentiation, apoptosis, and response to cellular stress. ERK1/2 extracellular signaling regulatory kinases are also part of the MAPK family. Overactivation of the ERK pathway appears as a frequent phenomenon in different types of cancer, including ovarian cancer.

The activation of MAPK (MAPK kinase kinase—MAP3K, MAPK kinase—MAPKK and MAPK) determines the activation in the cascade of ERK1 and ERK2 kinases, which is strongly conserved in evolution and controls cellular signaling in normal and pathological conditions. It has a greater role in angiogenesis and tumor metastasis than other signaling pathways.

C.The PI3K/AKT/MTOR signaling pathway includes, among others, proteins encoded by the PIK3CA, PTEN, STK11, MTOR, and AKT1 genes It represents one of the most important intracellular signaling pathways involved in various cellular activities, including cell cycle control, proliferation, protein synthesis, transcription, and angiogenesis. In ovarian cancer, the PI3K/MTOR pathway is the most frequently altered [61,62] (Figure 3).

Ovarian cancer is characterized by multiple changes in the proteins involved in the PI3K/AKT/MTOR signaling pathway through activation (in 65–75% of cases); thus, it is responsible for the increased invasion capacity of ovarian cancer. Ovarian neoplasm is an extremely heterogeneous group of neoplastic cells with diverse mutational phenotypes. Different ovarian cancer subtypes present patterns of different gene variants. The PI3K protein is a heterodimer with a p110 catalytic unit and a p85 regulatory unit. The activation, as an initial phenomenon to extracellular signals, of PI3K determines the activation/recruitment in the cascade of PDK1 and AKT.

AKT also has regulatory activity for many cellular processes as it is involved in survival and multiple metabolic processes.

One of the most important targets of AKT is represented by mTOR (serine/threonine kinase) with two components—mTORC1 and mTORC2—with functions in the control of cell growth. In most cases of ovarian cancer, mTOR is phosphorylated (hyperactivated).

D.The WNT/β-catenin signaling pathway is also involved in the regulation of various cellular processes, including cell proliferation, migration, angiogenesis, carcinogenesis, tumor progression, and treatment resistance (through aberrant activation). The WNT signaling pathway has three main subpathways—a classical/canonical WNT/ β-catenin/T cell factor pathway and two non-canonical pathways WNT/plenary cell polarity (PCP) and WNT/Ca2+ (Figure 4).

The classical pathway through aberrant activation determines the transition from epithelial to mesenchymal (EMT). The recurrence of metastasis and resistance to treatment appear in response to the triggered EMT process through the canonical signaling pathway, yet there are data that suggest that non-canonical pathways are also key processes of chemoresistance in mucinous ovarian cancer. The WNT signaling pathway thus represents a useful target for the identification of new therapeutic molecules.

Later, The Cancer Genome Atlas reported four similar subtypes, but without significant differences in prognosis or evolution.

E.JAK/STAT Pathway (Figure 5)

According to Bilal Rah et al. [63], the JAK/STAT signaling pathway is implicated in ovarian cancer. The significant importance of the JAK/STAT pathway in promoting tumor advancement and ensuring survival is widely recognized. Numerous studies have indicated that the aberrant activation of STATs strongly influences the expression of Bcl-2. Furthermore, elevated expression levels of cyclin D1 and c-Myc have been linked to the persistent activation of STAT3, thereby playing a role in promoting malignant transformations in human ovarian cancer.

The JAK–STAT signaling pathway consists of three main components: tyrosine kinase-associated receptor kinase, JAK tyrosine kinase, and STAT transcription factor, which is involved in various biological processes, such as cell division, differentiation, apoptosis, and immune regulation. The JAK–STAT signaling pathway is involved in the development of several tumors, such as gastric, breast, and colorectal cancers. In ovarian cancer, activation of the JAK–STAT signaling pathway is associated with neoplastic progression, so one of the treatment strategies for ovarian cancer may be considered the inhibition of the JAK–STAT signaling pathway [64,65,66].

F.The TP53 gene is prominently implicated in human cancer, particularly in high-grade serous ovarian cancer (HGSOC), where mutations occur at a frequency of at least 96%. Various TP53 mutations lead to the loss of wild-type functions, either due to the loss of DNA-binding activity or through a dominant-negative effect, wherein the mutated allele inhibits the function of the wild-type counterpart. Intriguingly, certain mutations exhibit gain-of-function properties independent of the wild-type TP53. Such gain-of-function mutations can enhance cell transformation and contribute to resistance against chemotherapy [67]. Additionally, the functional repercussions of TP53 mutations are contingent upon the specific mutation or its type. For instance, frameshift mutations are proposed to manifest distinct phenotypes compared to missense mutations. Notably, certain TP53 missense mutations generate full-length p53 proteins, often characterized by prolonged half-lives and the accumulation of inactive protein [68]. In contrast, frameshift mutations generally do not lead to p53 accumulation, and nonsense mutations typically yield unstable proteins.

TP53 mutations have been systematically classified based on their location, primarily within the DNA-binding domain (DBD), which is the most common site of aberrations. Additionally, mutations are categorized according to their oncogenic function, distinguishing between gain-of-function (GoF) and loss-of-function (LoF) mutations. Furthermore, they are classified by the type of mutation, such as missense, nonsense, frameshift, splice site, and indel mutations. Within missense mutations, subclassifications exist, including structural mutations suspected of affecting protein structure and activity, as well as functional mutations based on their ability to trans-activate promoters of p53 target genes. Notably, a subset of TP53 mutations has been identified as temperature sensitive, further expanding the spectrum of mutation characteristics under investigation [69].

TP53 unfavourable prognosis

A study has shown that, usually, TP53 LoF has been observed in type II tumors that are HGSOC (High Grade Serous Ovarian Carcinoma). The frequency of TP53 LoF had been observed “almost always,” which itself could be considered a negative prognosis of the pathology. As the authors have presented previously, type II tumors are much more aggressive compared to type I. Also, a particularity of the type II tumors is that they were found in advanced stages in over 75% of the cases they documented. They are usually formed in both ovaries, which is different from the type I tumors, which are usually singular, large, unilateral cystic neoplasms. Ascites is usually present in type II tumors due to the spread of the oncopathology in the omentum and mesenterium. Sadly, surgery and chemotherapy do not add up to the progression-free survival of the patients. This type of tumor accounts for roughly 90% of deaths due to ovarian cancer [70].

G.MET/HGF signaling pathway (Figure 6)

c-MET protooncogene, which is located at 7q31 locus of chromosome 7, encodes a membrane receptor involved in the tissue repair process and embryogenesis. Hepatocyte growth factor (HGF) represents the only known ligand for MET receptor. The binding determines c-Met activation through phosphorylation. The MET/HGF signaling pathway is involved in carcinogenesis through the following mechanisms: activation of oncogenic pathways RAS, STAT, PI3 K, β-catenin, and Notch; MMP expression, which leads to tumor progression and invasion; and the formation of new blood vessels. MET/HGF also interacts with other pathways, such as TGFα/EGFR [71,72,73].

MET/HGF is involved in the development of urogenital organs, ovarian surface epithelial transformation, proliferation, and invasiveness [74]. The cross-talk between MET/HGF and other pathways also plays a role in ovarian carcinogenesis. Examples are the cross-talk between RON and c-Met [75] and between the MET/HGF pathway and the EGF pathway [76].

H.Notch pathway (Figure 7)

The Notch signaling pathway plays a role in angiogenesis and cancer proliferation [77,78]. Notch represents the receptor in this highly conserved pathway [79]. The activating ligands for the Notch receptor are transmembrane proteins of the Delta and the Jagged/Serrate group: Jagged 1, Jagged 2 (JAG 1 and JAG 2), and Delta-like-1, -3, and -4 (DLL1, DLL3, and DLL4) [80]. E3 ubiquitin ligases are essential for the regulation of Notch signaling [81]. A complex cleavage process of the Notch receptor is initiated upon ligand interaction, which results in the release of the intracellular c-terminal domain, which translocates to the nucleus to interact with the transcriptional repressor CBF1/RBP-Jk [82]. CBF1 regulates the expression of a group of basic helix–loop–helix proteins encoded by HES genes [83].

Other signaling cascades that interact with the Notch pathway include the TGF-B, WNT, EGFR, HER2, and PI3K pathways [77].

The Notch pathway is involved in angiogenesis through the interaction with the VEGF pathway [84]. Epithelial ovarian cells express Notch ligands, especially JAG1, DLL1, and DLL4 [85].

I.EGFR

EGFR is overexpressed in the majority of epithelial ovarian cancers, and it is associated with poor outcome [86]. EGFR (ErbB-1 or HER1) is a transmembrane protein that functions as a receptor for extracellular protein ligands from the epidermal growth factor family, and it is a member of the ErbB family of receptors, which consists of EGFR (ErbB-1), HER2/neu (ErbB-2), HER3 (ErbB-2), and HER4 (ErbB-4) [87]. The location of the EGFR gene is on chromosome 7p12. EGFR signaling regulates cell growth, differentiation, motility, survival, and cancer.

EGFR family ligands include EGF- and EGF-like ligands, TGF-alpha, and heregulins (HRGs). Each ligand binds to a different member of the EGFR family. All EGFR family members prefer HER2 as a binding partner. HER3 is inactive on its own because it lacks intrinsic kinase activity. EGFR signals are transduced via intracellular adaptor proteins, which convey signals via cascades, such as the MAPK and PI3K/AKT pathways. The proteins further along in these signaling pathways have the ability to move from the cytoplasm into the nucleus. Within the nucleus, they communicate with transcription factors and associated complexes, including, but not limited to, MYC, ELK, and FOS/JUN [86,88].

J.RAS signaling pathway (Figure 8)

RAS represents a family of intrinsic GTP-binding proteins involved in signaling pathways that regulate a variety of cell responses, including proliferation, differentiation, migration, survival, and tumorigenesis [89,90]. RAS has three isoformes: HRAS, KRAS, and NRAS [91].

RAS represents a family of intrinsic GTP-binding proteins involved in various cellular signal transduction pathways that essentially regulate cell growth, differentiation, adhesion, and migration. The RAS subfamily is the most studied of the small G proteins due to its involvement in human tumorigenesis. RAS is one of the major pathways that has been most frequently found in several cancers, including pancreatic, lung, colorectal, ovarian, and hematopoietic cancers [91].

The role of the RAS, a protein subfamily in carcinogenesis, is well established. Approximately 20% to 30% of all tumors have mutations in one of the three RAS genes (KRAS, NRAS, and HRAS), with a relative frequency of involvement estimated at 85% for KRAS, 15% for NRAS, and 1% for HRAS, respectively [92].

## 6. Genetic Testing in Ovarian Cancer

Panel testing for hereditary breast and ovarian cancer

Genetic testing panels targeting high-penetrance ovarian cancer susceptibility genes are established as essential and medically justified for individuals with a personal or familial background of BRCA-associated cancer.

Genetic testing designed to assess hereditary cancer susceptibility plays a pivotal role in predicting an individual’s likelihood of developing cancer in the future. Research suggests that approximately 5–10% of all cancer cases are rooted in hereditary factors [93]. Typically, hereditary cancers manifest at a younger age and exhibit an autosomal dominant inheritance pattern within affected families.

Genetic testing is generally recommended when there is a personal or family history consistent with a hereditary susceptibility to cancer, the test can be adequately interpreted, and the results can be used to diagnose or influence the medical management of the at-risk person or family members [94].

The NCCN suggests that several specific genes may contribute to hereditary cancers, including but not limited the following list: BRCA1, BRCA2, BRIP1, CDH1, PALB2, PTEN, RAD51C, RAD51D, TP53, Lynch-syndrome-associated genes, and certain findings in ATM and CHEK2 [95].

In their 2020 study, Alvarado and colleagues conducted an evaluation involving 3162 women to determine the occurrence of pathogenic or likely pathogenic variants (PV/LPV) using a comprehensive multigene cancer panel encompassing 20 genes. A majority of these women (65.4%) had received diagnoses of breast or ovarian cancer. The study revealed an overall prevalence of 11.7% for any PV/LPV findings, with approximately 5.4% exhibiting BRCA1/2 mutations and 6.3% showing mutations in non-BRCA genes. When specifically considering the subgroup with PV/LPV results, it was noteworthy that 55% of all identified mutations were non-BRCA in nature. Consequently, the researchers concluded that employing multigene cancer panel testing might be a suitable approach for individuals in high-risk cohorts [96].

In 2020, Corredor and colleagues conducted an assessment of women who had experienced multiple primary breast cancers utilizing panel tests to determine the prevalence of non-BRCA mutations. A total of 85 women underwent testing with a multigene panel, and, among them, 33 individuals (38.8%) yielded positive results for pathogenic mutations. Specifically, these mutations included nine in BRCA1, five in BRCA2, five in ATM, one in BARD1, four in CHEK2, one in MSH2, one in MSH6, two in PALB2, one in PMS2, one in PTEN, and three in TP53. In summary, 17.6% of the tested individuals exhibited positive results for non-BRCA genes associated with predisposition to breast cancer [97].

In their 2020 publication, Daly et. al presented a comprehensive review of updates to the NCCN guidelines concerning screening for susceptibility to breast and ovarian cancer. They detailed alterations to the associated testing algorithms. These revised guidelines emphasize robust evidence supporting the notion that genes other than BRCA1/2, such as CDH1, PALB2, PTEN, and TP53, substantially elevate the risk of developing breast and/or ovarian cancer [98,99].

This change is of such significance that it necessitated renaming the page formerly known as “BRCA1/2 Testing Criteria” to “Testing Criteria for High Penetrance Breast and/or Ovarian Cancer Susceptibility Genes.” Furthermore, the testing criteria have been restructured into three distinct sections:Indications for clinical testing;Considerations for testing;Scenarios with low likelihood of clinical utility from testing.

The authors have also emphasized the potential consideration of multigene testing in cases where patients test negative for one specific syndrome yet their personal or familial medical history suggests the presence of another inherited cancer syndrome.

Additional noteworthy updates to the guidelines encompass revisions to the criteria governing the testing of individuals with Ashkenazi Jewish heritage and those related to pancreatic cancer screening.

Clinical utility of cancer predisposition genes

Determining if a cancer stems from an underlying CPG (cytosine–phosphate–guanine) mutation can have a profound effect on both the cancer patient and, potentially, their family members. In the term “CpG islands,” “CpG” stands for “cytosine–phosphate–guanine.” A CpG island is a region of DNA where a cytosine nucleotide is followed by a guanine nucleotide in the linear sequence of bases, which is connected by a phosphate group (CpG dinucleotide). CpG islands are often associated with gene regulatory regions in the genome. Consequently, CPG testing has been adopted as a standard practice for numerous genes, albeit typically in specific and carefully chosen instances [100].

Cancer diagnosis and management

Detecting a CPG mutation yields crucial insights that can significantly impact both diagnosis and treatment. Surgical approaches can be tailored accordingly, and they may potentially involve more extensive procedures when dealing with CPG mutation carriers, who face an elevated risk of future cancers. Adjustments to radiation therapy may be necessary, as certain CPGs are linked to heightened sensitivity to radiation. Similarly, chemotherapy strategies may be modified, with some treatments demonstrating greater efficacy while others prove less effective in individuals with CPG mutations. For instance, platinum-based therapies, which are not typically employed in breast cancer treatment, may find utility in BRCA1/2 carriers due to their heightened sensitivity to such drugs [101,102].

Emerging, personalized treatment approaches are continuously developed to directly address CPG mutations or exploit vulnerabilities in pathways triggered by CPG mutations. For instance, certain gastrointestinal tumors stem from inherited gain-of-function mutations in genes like KIT or PDGFRA, which could potentially be suppressed using drugs like imatinib [103]. Similarly, poly (ADP-ribose) polymerase (PARP) inhibitors that target DNA repair pathways, which become vulnerable in women with BRCA1 or BRCA2 mutations, were approved in the last decade [104].

Detecting an underlying CPG mutation can also offer valuable prognostic insights. For instance, survival rates vary significantly among different cancer types in CPG mutation carriers, with BRCA2-mutation-positive ovarian cancer patients experiencing significantly better outcomes and BRCA2-mutation-positive prostate cancer patients facing a less favorable prognosis. This underscores the importance of continuous assessment and personalized approaches to surveillance and risk reduction strategies for individuals with CPG mutations. Additionally, addressing non-cancer-related issues becomes crucial; for example, specific WT1 mutations can lead to gradual renal dysfunction, necessitating regular monitoring and early intervention.

Cancer screening and prevention

CPGs possess a unique quality, as they can function as predictive biomarkers for prospective diseases. Discovering a CPG mutation creates an opportunity to initiate proactive surveillance and risk reduction strategies aimed at minimizing or averting cancer development. The specific screening approach naturally depends on the cancer type, but it typically entails the use of imaging techniques to detect lesions before they become clinically evident. Preventive measures often involve surgical removal of vulnerable tissue, and they are typically reserved for non-essential organs in individuals at a very high risk, such as the stomach in CDH1 mutation carriers, the thyroid in RET mutation carriers, and the breasts and ovaries in BRCA1 mutation carriers [105].

Chemoprevention is an attractive strategy, but, to date, there has been little application. A notable exception is individuals at high risk for colorectal cancer in whom the risk of cancer is significantly reduced by daily aspirin [106].

While frequently underestimated, employing CPG mutation testing to identify relatives who do not carry the familial mutation holds significant benefits for both the patient and cost effectiveness. These individuals are relieved from personal and familial anxieties, thereby eliminating the need for costly screening and interventions.

Methods for testing cancer predisposition genes

DNA sequencing technology has evolved rapidly in recent years. Traditionally, gene testing has relied on developing individual assays for each CPG using expensive and time-consuming methods. It is now possible to sequence multiple genes in parallel, which is faster and cheaper than a single gene test with traditional methods, using a technique known as next-generation sequencing (NGS) [107].

As a result, this implies that a greater number of genes and a larger population may undergo genetic testing, especially when there is a diverse range of genes contributing to cancer susceptibility. Currently, gene panels comprising 5 to 100 cancer-predisposing genes are available, but it is likely that whole-genome sequencing will eventually replace them, allowing for the examination of genetic information related to various medical conditions and drug responses, as well.

Gene variants and mutations are common occurrences in our genome, and they are generally benign. Distinguishing between benign and pathogenic variants can be a challenging task, even for well-studied genes like BRCA1 that have been known for many years. Present-day genetic test reports are often intricate, and they can provide unclear information, such as the identification of variants with uncertain significance. Consequently, there is a lack of clear guidelines for managing these findings, leading to inconsistent, improvised, and sometimes inappropriate interventions. To achieve successful integration, it is crucial to involve expert genetic analysis and triage to establish clear protocols for management. While it is expected that all healthcare professionals should have a basic understanding of genomics, it is neither practical nor necessary for every clinician to become a genomics expert. Instead, the role and training of clinical geneticists need to be restructured to enable them to offer expert interpretations of the clinical implications of complex genetic tests, with the assistance of automated pipelines that generate routine results [105].

Tumor testing in ovarian cancer

The demand to expand the pool of ovarian cancer patients eligible for targeted therapies like PARP inhibitors has grown considerably. This has underscored the importance of tumor testing, which can uncover extra genetic mutations capable of predicting responsiveness to PARP inhibition. In fact, tumor testing can reveal an additional 4-7% of patients with pathogenic variants in BRCA1 and BRCA2, even if previous germline testing showed negative results for loss-of-function mutations. Furthermore, approximately 3% of these patients may exhibit somatic mutations in other genes associated with homologous recombination [108,109,110].

In theory, tumor testing has the capacity to identify both germline and somatic genetic changes, a capability lacking in germline testing alone. While the concept of relying solely on tumor testing may appear attractive, it is essential to acknowledge several factors that could impact the precision of such tests, making it less reliable as a definitive method for detecting germline variants. Notably, in the majority of instances, tumor testing relies on DNA extracted from formalin-fixed paraffin-embedded (FFPE) tissues, which can pose technical challenges in terms of amplification and obtaining precise results.

To secure tumor DNA, it is imperative to perform microdissection of the tumor, making small diagnostic biopsies impractical for subsequent analysis. It is crucial to recognize that weak binding can result in DNA fragmentation, and the use of formalin can induce C>U deamination, potentially causing sequencing errors and the potential for inaccurate mutation identification. Additionally, one must consider the tumor’s heterogeneity and the precise proportion of tumor cells within the specimen from which the DNA for analysis will be extracted. Lastly, it is essential to acknowledge and account for the limitations inherent in these methods.

Identifying significant genomic rearrangements or pathogenic variants (PVs) in genomic regions associated with gastric cancer (GC) can be particularly challenging when working with DNA from formalin-fixed paraffin-embedded (FFPE) samples [111]. Additionally, it is worth noting that genetic reversion of the initial BRCA1 or BRCA2 variant can occur, often as a consequence of chemotherapy or as an adaptive resistance mechanism [112]. Considering these factors, while tumor testing holds valuable implications for treatment decisions, it cannot definitively substitute traditional germline testing for evaluating hereditary cancer predisposition.

Testing for repair deficiencies through homologous recombination

It is clear that most ovarian tumors exhibiting homologous recombination (HR) deficiency originate in individuals carrying germline pathogenic variants (PVs) in BRCA1 and BRCA2 genes. Nonetheless, some tumors may experience HR pathway impairment due to alternative genetic anomalies, which can manifest in either the germline or the somatic cells, as mentioned earlier. In fact, according to information from the TCGA (The Cancer Genome Atlas), as many as half of high-grade ovarian cancers contain PVs in genes associated with the HR pathway [108]. Of course, it should be clarified that not all genetic changes in these genes will lead to HR deficiency.

Identification of HR deficiency

The recognition of HR deficiency in a tumor can be assessed through supplementary examinations capable of gauging genomic instability. These tests generate an HR deficiency score by assessing generic loss of heterozygosity (LOH), telomeric allelic imbalance, and large-scale state transitions. Functional assays of this kind aim to identify the genomic alterations that result from the accumulation of genetic changes, and they are predominantly developed and patented by commercial entities. Both the LOH deficiency score and HR have been adopted as biomarkers in numerous studies evaluating the effectiveness of different PARP inhibitors, such as niraparib [113] and rucaparib [114].

Notably, in the recent ARIEL2 study, 34% of BRCA wild-type patients initially categorized as having low LOH based on archival specimens showed a shift to high LOH status in their pretreatment specimens, indicating a change in biomarker status over time. Conversely, no instances were observed where the classification shifted from high LOH to low LOH between the archival and pretreatment specimens. Given the diverse and intricate genetic events in ovarian tumors, it is advisable to employ a combination of tests to enhance accuracy.

Recent advances in ovarian cancer therapeutics have underscored the growing demand for genetic testing among patients. Primarily, the focus has been on somatic or tumor testing to gauge the suitability of PARP inhibitor treatment. Tumor genomic testing has the capability to detect both germline mutations (typically with a variant allele frequency (VAF) close to 50% or higher) and somatic mutations (VAF>5%). However, the quality of tumor DNA can be compromised when preserved in paraffin, potentially hindering the feasibility of comprehensive molecular assessments. Additionally, paraffin preservation can lead to cytosine deamination, introducing sequencing errors as a result.

HRD testing

Homologous DNA damage repair system deficiency can arise from either germline or somatic mutations in the relevant repair genes (notably, BRCA1/2). Around 50% of high-grade ovarian cancer patients exhibit homologous recombination deficiency due to pathogenic or potentially pathogenic mutations in these repair genes.

Homologous Recombination Deficiency (HRD) testing is a comprehensive assessment of genetic and genomic factors that collectively indicate whether cancer cells have defects in their homologous recombination repair mechanisms. These defects make cancer cells more susceptible to certain treatments, such as PARP inhibitors and platinum-based chemotherapy. HRD testing incorporates certain factors, such as the genetic mutation identified. The most common mutations tested for are in the BRCA1 and BRCA2 genes, but other genes may also be considered. The most important factors to incorporate in the calculation of the genomic instability are:Loss of heterozygosity (LOH), a genomic event in which one allele of a gene is deleted or inactivated, leaving only one functional copy.Telomeric allelic imbalance (TAI), which refers to imbalances in the lengths of the telomeres (protective caps at the ends of chromosomes) between two homologous chromosomes. TAI can be indicative of HRD, and it is evaluated in HRD testing.Large-scale state transitions (LST), which measure the number of chromosomal breaks, gains, or losses in a cancer genome. A high LST score is associated with HRD.Microhomology-mediated end joining (MMEJ), which is a DNA repair mechanism that can be more active in HRD cells. It involves the repair of double-strand DNA breaks by joining DNA ends with short, overlapping sequences.Depletion of RAD51 foci, which is a reduction in the formation protein involved in homologous recombination repair.

The combination of these factors helps tumor boards as well as multi-disciplinary teams determine whether a patient’s cancer cells are HRD positive or HRD negative. HRD-positive tumors are more likely to respond to treatments that exploit HRD, such as PARP inhibitors or platinum-based chemotherapy.

It is important to note that the specific tests and methods used for HRD testing may vary between laboratories and may continue to evolve with ongoing research; this is why we do not describe a particular type of test on the current market. The results of HRD testing play a crucial role in personalized cancer treatment decisions and in identifying the most appropriate therapeutic strategies for individual patients.

The molecular assessment for identifying homologous recombination deficiency involves examining genomic instability markers, such as loss of heterozygosity, telomeric imbalance, large-scale state transitions, microhomology-mediated end joining, or depletion of Rad51 Foci. Genomic analyses have demonstrated that patients with homologous repair system deficiency exhibit a more favorable response to PARP inhibitors (PARPI). Furthermore, each cancer type is associated with distinct genomic characteristics, highlighting the need for tailored approaches.

Genetic testing is advised for all ovarian cancer patients. While family history can be significant, it does not consistently predict genetic mutations. A study conducted by Song et al. [115] on individuals with ovarian neoplasms revealed that roughly 47% of patients with germline mutations did not have a documented family history of such mutations.

Genetic testing plays a pivotal role in tailoring the treatment approach for each patient, while recognizing that those with BRCA mutations tend to exhibit heightened responsiveness to platinum-based chemotherapy and targeted therapies.

## Figures and Tables

**Figure 1 ijms-24-15987-f001:**
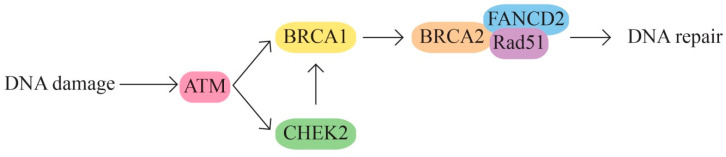
BRCA1 and BRCA2 pathway.

**Figure 2 ijms-24-15987-f002:**
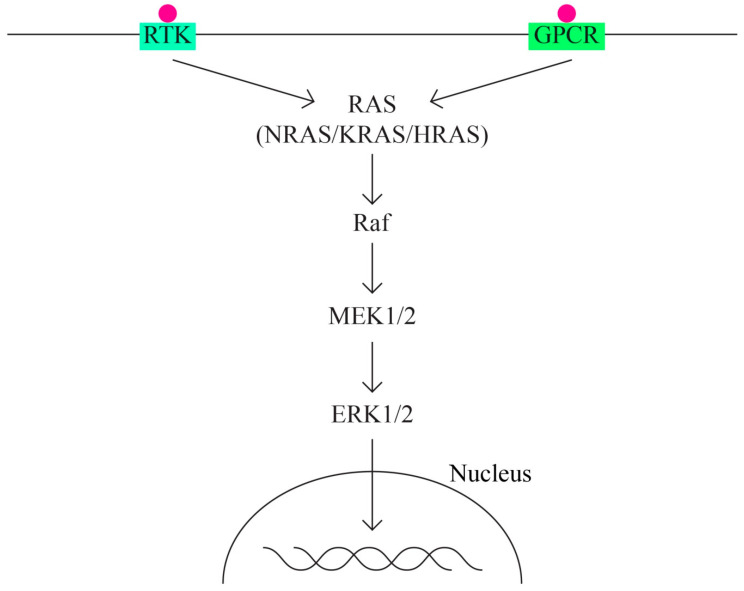
MAPK/ERK pathway.

**Figure 3 ijms-24-15987-f003:**
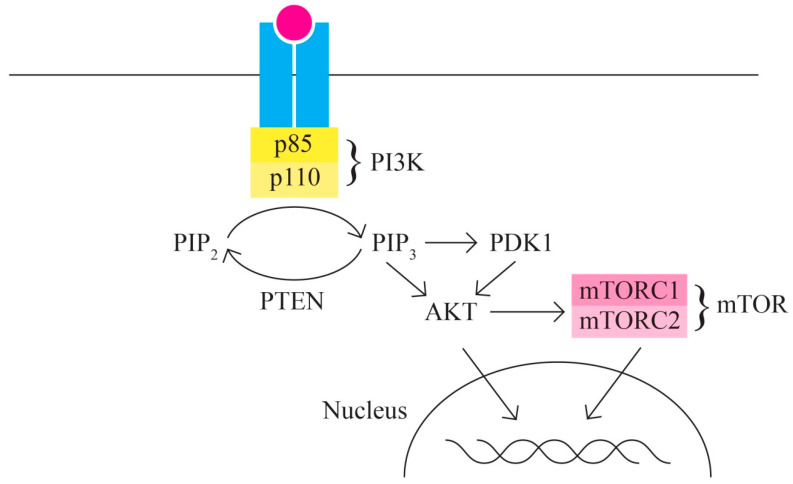
PI3K/AKT/MTOR signaling pathway.

**Figure 4 ijms-24-15987-f004:**
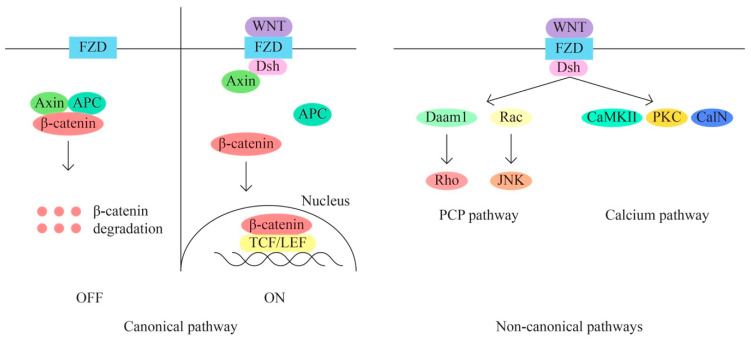
WNT/β-catenin signaling pathway.

**Figure 5 ijms-24-15987-f005:**
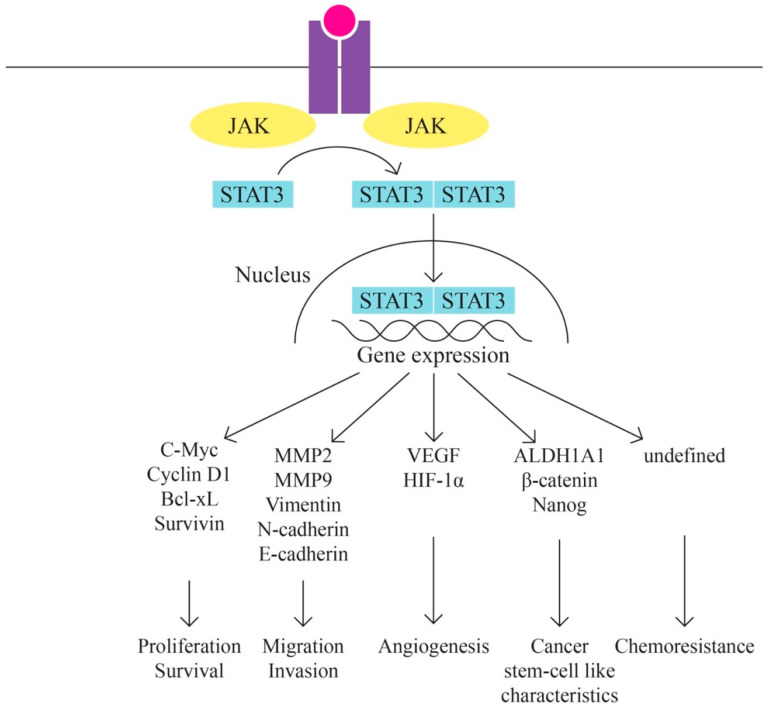
JAK/STAT pathway.

**Figure 6 ijms-24-15987-f006:**
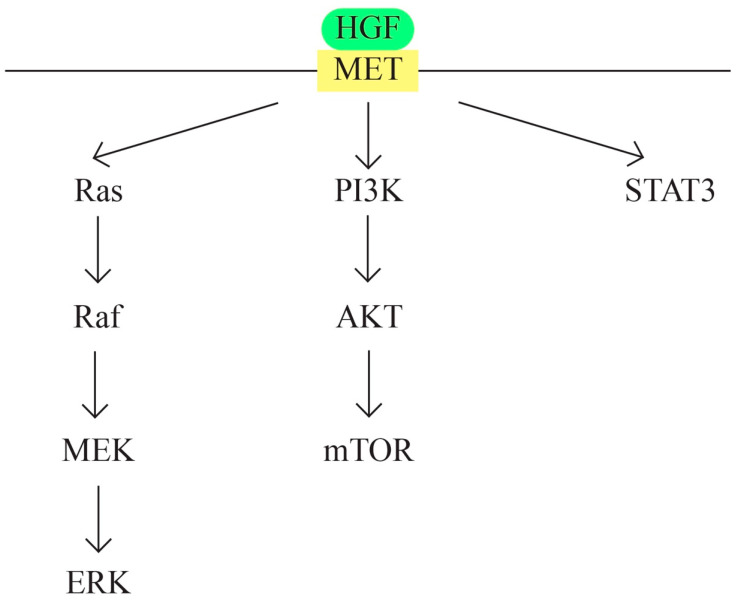
MET/HGF signaling pathway.

**Figure 7 ijms-24-15987-f007:**
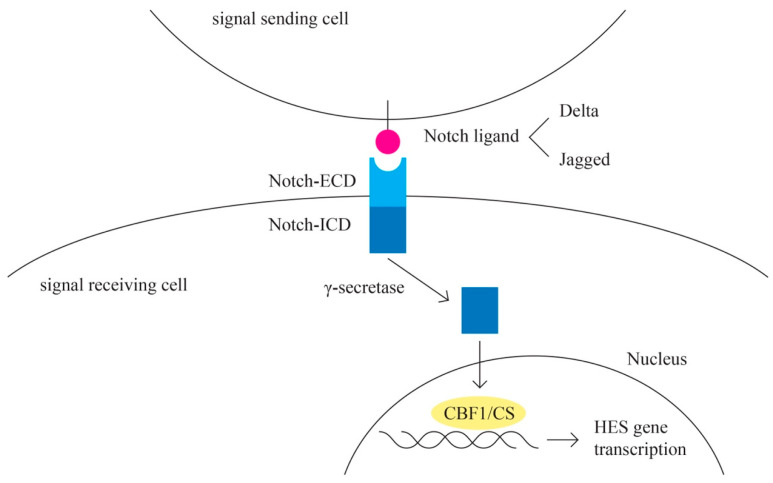
Notch pathway.

**Figure 8 ijms-24-15987-f008:**
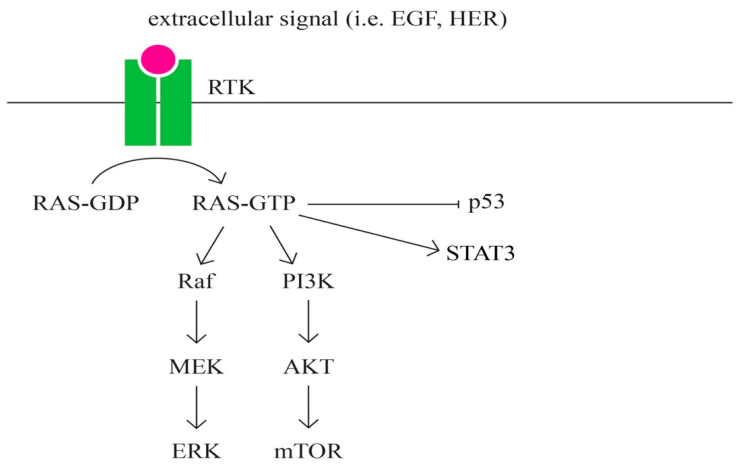
RAS signaling pathway.

**Table 1 ijms-24-15987-t001:** Subtypes of ovarian cancer and frequency of genetic alterations [10].

Subtype	Frequency of Genetic Alterations [10]			
	TP53	BRCA 1/2	PIK3CA	KRAS
HGSOC	96%	22–40%	2.9%	5.9%
LGSOC	8.3	10%	12.5%	54%
EnOC	5–54.5%	11.1%	31.4%	10.3%
OCCC	10%	4.5%	51%	15%
Mucinous	56.8%	0	13.5%	57.1–64.9%

**Abbreviations:** HGSOC, high-grade serous ovarian carcinoma; LGSOC—low-grade ovarian serous carcinoma; EnOC, endometrioid ovarian carcinoma; OCCC, ovarian clear cell carcinoma.

## Data Availability

If needed, we will find a way to provide further information.

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
