# Peer review of "Developments in Genetics: Better Management of Ovarian Cancer Patients"

_ijms, 2023, doi:10.3390/ijms242115987_

Round 1

Reviewer 1 Report

Comments and Suggestions for Authors

The manuscript by Maioru et al. aimed to describe the molecular feature that basically guide the management of ovarian cancers. Basically the subject could be of interest to summarize this field to the scientific readers especially in the context of the new technical approaches that allow the emergence of novel therapeutic targets or better therapeutic approaches.

However the manuscript could be better organize especially taking into account recent publications

Please, see below a couple of them:

Kostantinopoulos et al.. Nature Cancer 2023, or J Clin Onc 2022 2020.

Below please find other comments/concerns.

-          Abstract need to be reworded in lines 13-15.

-          Please, try to avoid repetitions and edit those paragraph reporting them.  Paragraph 2 and 3: there are several repetitions, i.e.34-38 and 39-43 or lanes 84-90 as well as 92-94, those concepts are repeated in the same paragraph or in other paragraphs.

-          Information about the specific genes. Instead to report their role in normal cells, it would mandatory to report the alternative function in ovarian cancers. For example for p53 the type of mutations are missed and therefore their functions and clinical impact. BRCA 1 and 2 are mentioned in several paragraphs with a lot of repetitions. Among the relevant genes, ARID1 is missed. What about the impact of Lynch syndrome on ovarian cancer onset? Please complete all the above mentioned suggestions taking into account the publications cited above,.

-          Lanes 191-199, references are missed.

-          Paragraph 5, about the signalling pathways. As for the genes the mentioned signalling pathways might be altered in ovarian cancers for specific reason, i.e. mutation of a gene, over-expression and so on. The mere list of these pathways is useless in this context as it is.

-          The description of some abbreviation is missed, i.e. CPG, HRD and so on.

-          For the NCCN guidelines you might refer to Armstrong D.K. 2022.

-          The HRD could be presented better taking into account how important is at the moment for therapeutic decision. There are description about BCRA genes ones again but, for example, the types of molecular tests are not really reported and the new advances on this side. Please consider to enlarge the information on this field with the new publications of the last two years.

Comments on the Quality of English Language

Minor editing of English is required. Pay attention to the abstract also

Author Response

We have written our answers in the red background color. Dear reviewer, thank you for your precious time spent on making our manuscript better! 

          Abstract need to be reworded in lines 13-15.

Which particular part do you consider should be reworded?

-          Please, try to avoid repetitions and edit those paragraphs reporting them.  Paragraph 2 and 3: there are several repetitions, i.e.34-38 and 39-43 or lanes 84-90 as well as 92-94, those concepts are repeated in the same paragraph or in other paragraphs.

We have cut the paragraphs which seemed to repeat itself. Thank you for your insight!

-          Information about the specific genes. Instead to report their role in normal cells, it would mandatory to report the alternative function in ovarian cancers. For example for p53 the type of mutations are missed and therefore their functions and clinical impact. BRCA 1 and 2 are mentioned in several paragraphs with a lot of repetitions. Among the relevant genes, ARID1 is missed. What about the impact of Lynch syndrome on ovarian cancer onset? Please complete all the above mentioned suggestions taking into account the publications cited above,.

We have included both the p53 mutations and the ARID1A gene you suggested. Thank you for your great suggestions!

-          Lanes 191-199, references are missed.

This is pure speculation on our part; we do not consider for a fact that these tumors have a particular genetic pattern, we only suggest they might until further research has been done. This is why there is no reference there. Sorry for the confusion.

-          Paragraph 5, about the signaling pathways. As for the genes the mentioned signalling pathways might be altered in ovarian cancers for specific reason, i.e. mutation of a gene, over-expression and so on. The mere list of these pathways is useless in this context as it is.

In paragraph 5, we present the genes and their function as well as the oncogenesis process. Discussing all the possible mutations or solely pathogenic mutations as well as over-expression in various cells would be a huge amount to cover.

-          The description of some abbreviation is missed, i.e. CPG, HRD and so on.

We have corrected this and have added new information regarding HRD and characteristics about it! Thank you!

-          For the NCCN guidelines you might refer to Armstrong D.K. 2022.

We have referred Armstrong D.K. 2022, as per your suggestion, we apologize for this major oversight!

-          The HRD could be presented better taking into account how important is at the moment for therapeutic decision. There are description about BCRA genes ones again but, for example, the types of molecular tests are not really reported and the new advances on this side. Please consider to enlarge the information on this field with the new publications of the last two years.

We are currently working on another article regarding the efficacy of HRD treatment and PARP inhibitors, this is the reason we have not expanded on this particular side.

Reviewer 2 Report

Comments and Suggestions for Authors

In this review, the authors described global ovarian cancer incidence and survival, risk factors, genetic findings in epithelial and mesenchymal ovarian tumors, molecular pathways implicated in ovarian cancer, and genetic tests.

The review described distinct genetic markers in various ovarian tumor subtypes, including epithelial and mesenchymal tumors. The genetic makeup helps in classification of ovarian tumors, shedding light in histomorphology with underlying genetic alterations. Distinct genetic alterations are helpful in the diagnosis of ovarian tumors, especially in difficult cases.  

The authors listed genetic causes of ovarian oncogenesis processes, described in details of genes implicated in various ovarian cancer, their genetic characteristics, mechanisms, and most common genetic alterations of the genes, including candidate genes as well. 

Multiple molecular pathways in ovarian cancer are introduced, with brief figures for explanation of the pathways, making it easy for readers to understand.

Germline testing for hereditary ovarian cancer and genetic testing for cancer predisposition genes were described in details. Homologous recombination deficiency (HRD) has been found to be clinically significant in ovarian cancer management, which  indicated a more favorable response to PARP inhibitors. There is no standard test for HRD, but there are several commercial tests available. Hope the authors will provide more details of the HRD testing.  

Abbreviation is used frequently in the review. Please indicate what CPG stands for. 

Author Response

Thank you for your kind words on our manuscript. We have corrected and changed some of the repeated abbreviations and have also explained what CpG islands are. Thank you for your precious time to review our work!

Reviewer 3 Report

Comments and Suggestions for Authors

I read with great interest the manuscript, which falls within the aim of this Journal and offers a high-quality overview of the topic that is of primary importance.
The abstract perfectly summarizes the contents of the manuscript and the figures are clear and interesting.

Although the manuscript can be considered already of high quality, I would suggest taking into account the following minor recommendations:

- I suggest another round of language revision, in order to correct a few typos and improve readability.

- Considering the topic analyzed and state of the art in literature, the authors could extend and improve the introduction by evaluating and citing current evidence about possible other target therapeutic strategies for patients with ovarian cancer.  I would be glad if the authors discussed this important point, referring to PMID: 37314974.

The presentation of the manuscript is complete and interesting and in my opinion, it is scientifically sound and offers a significant contribution to the field, addressing future research priorities. For all these reasons, I recommend the publication of the article, pending few minor revisions.

Comments on the Quality of English Language

Minor editing of the English language is required to make the work clearer and more readable.

Author Response

Thank you for your kind words on our manuscript. We have corrected and changed some of the repeated abbreviations, looked for passages in the manuscript where the readability could be considered ambiguous and have also explained what CpG islands are.

We are currently working on a further study regarding the therapeutic benefit offered by the treatment with PARP inhibitors in the patients who get genetic testing in regards to BRCA 1/2, HRD status in our facility. Since the patients are not ours and only the testing is done in the facility, it will take some time until we publish the data we are currently gathering. We love to see this high interest in this particular type of testing

 Thank you for your precious time to review our work!